# Molecular Players of EF-hand Containing Calcium Signaling Event in Plants

**DOI:** 10.3390/ijms20061476

**Published:** 2019-03-23

**Authors:** Tapan Kumar Mohanta, Dhananjay Yadav, Abdul Latif Khan, Abeer Hashem, Elsayed Fathi Abd_Allah, Ahmed Al-Harrasi

**Affiliations:** 1Natural and Medical Sciences Research Center, University of Nizwa, Nizwa 616, Oman; abdullatiff@unizwa.edu.om; 2Department of Medical Biotechnology, Yeungnam University, Gyeongsan 38541, Korea; dhanyadav16481@gmail.com; 3Botany and Microbiology Department, College of Science, King Saud University, Riyadh 11451, Saudi Arabia; habeer@ksu.edu.sa; 4Mycology and Plant Survey Department, Plant Pathology Research Institute, ARC, Giza 12511, Egypt; 5Plant Production Department, College of Food and Agriculture Science, King Saud University, Riyadh 11451, Saudi Arabia; eabdallah@ksu.edu.sa

**Keywords:** calmodulin, calmodulin-likes, calcium dependent protein kinases, calcineurin B-like, EF-hands

## Abstract

Ca^2+^ is a universal second messenger that plays a pivotal role in diverse signaling mechanisms in almost all life forms. Since the evolution of life from an aquatic to a terrestrial environment, Ca^2+^ signaling systems have expanded and diversified enormously. Although there are several Ca^2+^ sensing molecules found in a cell, EF-hand containing proteins play a principal role in calcium signaling event in plants. The major EF-hand containing proteins are calmodulins (CaMs), calmodulin like proteins (CMLs), calcineurin B-like (CBL) and calcium dependent protein kinases (CDPKs/CPKs). CaMs and CPKs contain calcium binding conserved D-x-D motifs in their EF-hands (one motif in each EF-hand) whereas CMLs contain a D-x_3_-D motif in the first and second EF-hands that bind the calcium ion. Calcium signaling proteins form a complex interactome network with their target proteins. The CMLs are the most primitive calcium binding proteins. During the course of evolution, CMLs are evolved into CaMs and subsequently the CaMs appear to have merged with protein kinase molecules to give rise to calcium dependent protein kinases with distinct and multiple new functions. Ca^2+^ signaling molecules have evolved in a lineage specific manner with several of the calcium signaling genes being lost in the monocot lineage.

## 1. Introduction

Calcium is one of the most important plant nutrients that plays diverse roles in plant growth and development, as well as stress responses. It is a divalent cation that is obtained from the soil through the root system of the plants. Calcium is an essential macronutrient for plants [1]. The Ca^2+^ uptake from the soil occurs through the apoplast or symplast and is deliver to the above ground shoot through the xylem [2]. The Ca^2+^ is an essential nutrient for the normal development of the plant root and shoot tips and more specifically in cell division. It is also involved in the formation of microtubules, that play a critical role in the separation of chromosomes during the anaphase of the cell division [3,4,5,6]. Calcium pectate is found in the interface between the cell walls and provides structural stability to the cell and cell membrane [7]. Ca^2+^ accumulates extracellularly as calcium pectate and bind the neighboring cells together. Ca^2+^ is a versatile second messenger involved in signal transduction and regulates diverse physiological processes, such gene expression, ion balance, as well as carbohydrate, lipid and protein metabolism.

Several environmental stimuli and stress factors can modulate the free cytosolic Ca^2+^ concentration (100 nM–1000 nM) that has impact on the growth and development of the plants [8,9]. Different plant species vary in their Ca^2+^ requirement where monocot plants requiring less Ca^2+^ than the dicots for their growth and development [10]. In response to the external stimuli, the concentration of extracellular Ca^2+^ increases, leading to an increase in cytosolic Ca^2+^ whose distribution is balanced by the Ca^2+^ chelating proteins and sub-cellular organelles like vacuoles, mitochondria, endoplasmic reticulum (ER), as well as the cell wall. Cellular organelles enclosed with a double membrane also capable of generating a Ca^2+^ signal on their own [11]. The cytosolic concentration of free Ca^2+^ in a resting cell is approximately 10^−7.5^ M and can rise approximately ten-fold in response to an external stimulus. However, cytosolic concentration above 10^-7^ M is cytotoxic and deleterious to the cell function [12,13,14]. The Ca^2+^ level within the ER is maintained at a concentration that is at least 10–15 times higher than the level of cytosolic Ca^2+^ [15]. Plastids and mitochondria contain micromolar concentrations of Ca^2+^, which is comparatively less than the levels found in the ER.

However, studies have reported that the Ca^2+^ signaling event of sub-cellular organelles are independent of cytosolic Ca^2+^ levels [16,17]. The sieve tubes (cells lack nucleus at maturity) in vascular tissues contain 20–100-fold higher level of Ca^2+^ relative to other plant cells [18]. Environmental stimuli and hormonal signals can rapidly and strongly modulate cytosolic Ca^2+^ concentration in a transient manner. Thus, appropriate regulation of cytosolic Ca^2+^ levels is crucial for cell survival. Exorbitant cytosolic Ca^2+^ levels can be cytotoxic and lead to the cell death. The presence of chelating and buffering calcium-binding proteins in the cytosol used to moderate and balance free calcium level and facilitate the localization and spatial distribution of Ca^2+^ ions. Although external stimuli modulate cellular Ca^2+^ level, Ca^2+^ signals are also defined by a variety of kinetic parameters, such as origin, localization, time, duration, frequency and amplitude of the stimulus. The kinetics of Ca^2+^ signals are largely dependent on the strength of the external stimulus. Ca^2+^ signals are also cell-, tissue- and organ-specific. Ca^2+^ signatures regulate specific responses that allow plants to adapt to the prevailing conditions. Notably, elevations in Ca^2+^ concentration in cellular compartment may regulate different responses depending upon the nature of the stimulus.

Plants encode different types of calcium-binding proteins that are used to chelate and buffer the cytosolic Ca^2+^ to avoid the cytotoxicity. These proteins are classified as EF-hand-containing sensor protein/protein kinases, non-EF hand-containing Ca^2+^ binding proteins and transporter/pump proteins. The calcium transporter/pumps actively transport Ca^2+^ outside of the cytosol to maintain appropriate cytosolic Ca^2+^ levels. Ca^2+^-ATPase, P-type ATPase IIA family, P-type ATPase IIB family and Ca^2+^/H^+^ antiporters are the major Ca^2+^ transporter/pump proteins. The EF-hand-containing proteins actively bind to Ca^2+^ and chelate the cytosolic calcium to regulate calcium homeostasis. The EF-hand-containing calcium-binding proteins are calmodulins (CaMs) [19], calmodulin-like (CMLs) [19], calcineurin B-like (CBL) [20], calcium-dependent protein kinases (CDPK/CPKs) [21], CPK-related protein kinases (CRKs) and calcium and calmodulin dependent protein kinases (CCaMKs). There are also several proteins present in the cell that do not possess an EF-hand but still bind Ca^2+^. These proteins are phosphoenolpyruvate carboxylase kinase-related kinases (PEPRKs), annexins, calnexin, calreticulin, forisomes, pistil-expressed Ca^2+^-binding proteins and phospholipase D. Although functional studies of these proteins have been conducted, many detail and genomic aspects are still lacking. More specifically, the major signature sequences responsible for the binding Ca^2+^ has not been elucidated in Ca^2+^ transporter/pumps and non-EF-hand containing proteins. In addition, it is also important to understand the interacting network of calcium signaling proteins. This manuscript has reviewed the complex interactome network and genomic aspects of calcium binding signature sequences of EF-hand containing proteins, CaMs, CMLs, CBLs and CPKs. In the review, detailed genomic, evolutionary and complex interactome network of important EF-hand containing calcium binding proteins are presented and discussed.

## 2. Calmodulins (CaMs)

The genome of various plant species encodes 1 to 13 *CaMs* with the number of *CaMs* being independent of genome size (Table 1). For example, the genome size of the algal species, *Osterococcus lucimarinus* is 13.2 Mb and contains only two *CaMs*, whereas the genome size of *Eucalyptus grandis* is 691 Mb and encodes only one *CaM* (Table 1) [19]. The genomes of the algal species, *Coccomyxa subellipsoidea* and *Chlamydomonas reinhardtii,* encode three and six *CaMs*, respectively. The genomes of wild mustard, *Brassica rapa* and the common monkey flower, *Mimulus guttatus* contain 13 *CaMs.* The average number of CaM proteins encoded per plant species is 6.6 with most plant species encoding less than 10 *CaMs* in their genome [19]. The largest and smallest *CaM* genes reported to date in the plant kingdom are *PvCaM1-9* (*Panicum virgatum CaM*, 1314 nucleotides CDS) and *AcCaM5-3* (*Aquilegia coerulea CaM*, 399 nucleotides CDS), respectively. The majority of *CaMs* contain introns with only 5.16% of the *CaMs* are intronless [19]. A study of the *CaM* gene family in 41 plant species reported that 41.69%, 12.91%, 31.73%, 2.21%, 1.84% and 2.85% of the *CaM* genes contain one, two, three, four, five and six introns, respectively [19]. No *CaM* genes have been reported to contain seven or more introns.

CaMs are small globular protein containing N- and C-terminal EF-hand pairs. The size of the CaM proteins are ranges from 124 (CsCaM4) to 438 amino acids (PvCaM1–9). The isoelectric points of CaM proteins are range from 3.95 (MgCaM9-2) to 6.279 (CsubCaM5-2) (Appendix A). Upon binding of Ca^2+^ to the EF-hands, the globular structure of the EF-hand is modified into an open conformation that allow the protein to interact with other proteins [22,23]. These interactions lead to the activation or repression of target proteins, thus translating Ca^2+^ signals into a biochemical response [24,25]. The EF-hands of CaMs contain 12 amino acids in a canonical helix-loop-helix structure. Therefore, each EF-hand contains 36 amino acids. Ca^2+^ bind to the EF-hands in a pentagonal bipyramidal geometry with seven coordination sites [21,26]. In the majority of cases, Ca^2+^ is coordinated through the side chain oxygen at X, Y and Z, by a water molecule at –X, by a backbone carbonyl at -Y and by a carboxylate molecule at –Z vertices of the EF-hands, making it a bidentate interaction with the metal ion [26,27]. In position X7, Ca^2+^ bind to the amino acids via a hydrogen-bonded water molecule (Figure 1). Positions X and Y are always occupied by either Asp (D) or Asn (N); Z is occupied by D, N or serine (S); -Y contains any amino acid; whereas –Z contribute two coordinate sites that contain a Glu (E) [26]. However, frequent substitution has been observed at position –Z where Glu can be substituted by Asp (D). The CaM proteins contain a conserved Ca^2+^ binding D-x-D signature signal sequence in the EF-hand domain (Table 2) [19]. The conserved Asp amino acid of D-x-D motifs are present coherently and each EF-hand contain one D-x-D motif [19]. Along with the side chain oxygen and back-bone carbonyl atom, the conserved Asp amino acids of D-x-D motifs are responsible for binding Ca^2+^ in the EF-hand (Figure 1).

The D-x-D motifs are conserved at the 14th and 16th position of each EF-hand whereas the 15^th^ position is occupied by any amino acid [19]. In addition to the D-x-D motifs, the first EF-hand contains conserved an E-x_2_-E motif at the 5th and 8th position and an E amino acid at the 25th position [19]. The E amino acid in the 2nd EF-hand is conserved at 5th and 12th position, at the 22nd, 23rd, 25th and 26th position of the D-F-x-E-F motif and also has a conserved D amino acid at the 36^th^ position [19]. The third EF-hand contains a conserved D amino acid at the 1st position and an E amino acid at 8^th^, 25th and 36th position [19]. The 4th EF-hand contains conserved E amino acid at the 4th, 5th, 12th and 25th position [19]. Conservation of the E amino acid is observed at the 5th position of the 1st, 2nd and 4th EF-hand, while conservation of E-amino acids is observed at the 25th position in all of the four EF-hands [19]. The CaMs bind to their target proteins in CaM binding domains (a stretch of 16–35 amino acids in the target protein) present in the target protein [29,30]. In few instances, CaM bind to its target protein via an IQ motif, in a Ca^2+^ dependent manner. The IQ motif contains a conserved consensus sequence of I-Q-x_3_-K/R-G-x_3_-R amino acids where the I amino acid can sometimes be substituted by a F/L/V amino acid [22,30]. The sequence alignment analysis indicate that CaMs are highly conserved throughout their protein sequence [19]. The presence of multiple genes encoding identical or nearly identical proteins represent an interesting aspect of evolution. Duplicated genes gradually acquire mutations that often lead to a divergence in the coded sequence and closely-related paralogs lead to the functional divergence. Duplicated genes that do not undergo any mutations, remain under selective pressure to maintain the original protein sequence and function. Interestingly, CaMs are believed to be under strong selection pressure to maintain the original protein sequence intact [26].

Post-translational events play a crucial role in protein signaling, trafficking, sub-cellular localization, metabolism and regulation. Palmitoylation and myristoylation are two post-translational processes that occur in calcium signaling proteins where they direct the sub-cellular localization of the protein. CaM proteins do not contain any signaling sequences for palmitoylation or myristoylation event [19]. Therefore, CaM proteins are always found in the cytosol and function in binding Ca^2+^. The major reason for the high level of sequence conservation in CaMs may be because CaMs are required at levels that exceed single gene output. In this case, concurrent production from multiple genes would be an imperative. Alternatively, different proteins may have evolved specific regulatory responses or expression patterns. Such divergence and specialization would favor the selection and retention of paralogs.

## 3. Calmodulin-Like (CMLs)

A genome-wide analysis of the *CML* gene family across 40 plant species indicated the presence of a variable number of *CML* genes in each genome. The number of *CML* genes present in a plant genome is independent of the genome size. The number of *CML* genes in plant genome ranges from 2 to 47 till the date studied so far (Table 1) [19]. *Coccomyxa subellipsoidea* and *Ostreococcus lucimarinus* possess only two *CML* genes, whereas *A. thaliana* possesses forty-seven *CMLs*. The monocot plants, *Brachypodium distachyon*, *Oryza sativa*, *Panicum virgatum, Setaria italica, Sorghum bicolor and Zea mays* contain 23, 33, 20, 17, 22 and 21 *CMLs*, respectively (Table 1) [19]. The dicot plants, *Brassica rapa, Eucalyptus grandis, Gossypium raimondii, Solanum tuberosum and Vitis vinifera* contain 36, 25, 30, 27 and 13 *CMLs*, respectively (Table 1) [19]. Analysis of the individual *CML* genes indicates that the majority of *CMLs* are intronless, suggesting their ancient origin. Approximately, 71.72% of the *CMLs* are intronless whereas 9.5% have one, 2.88% have two, 5.29% have three, 3.48% have four and 1.8% have five introns [19]. Only a small number of *CMLs* contain six, seven, eight or nine introns, where no *CMLs* contain more than ten introns [19]. Although the majority of *CMLs* are intronless, the presence of introns in a few *CML* genes suggest that these introns have evolved recently. The observation that 71.72% *CML* genes are intronless suggests that *CML* genes are highly conserved orthologous genes and have evolved from a common ancestor.

CML proteins are also contain four calcium-binding EF-hands (Figure 2) that share at least 16% sequence similarity with each other [31]. The majority of CMLs, however, share less than 50% sequence similarity with CaMs [32]. Most of the CMLs contain 100–150 more amino acids in their proteins relative to the CaMs. Although there is considerable sequence divergence exist between CaMs and CMLs, still CMLs have the capacity to bind Ca^2+^. The size of CML proteins range from 115 (AcCML25-3) to 703 amino acids (RcCML23). The isoelectric points of CML proteins are range from 3.263 (PtMCL25-3) to 9.703 (PhCML11) (Appendix A). The CMLs exhibit a shift in electrophoretic mobility in the presence of Ca^2+^, suggesting that they function as important Ca^2+^ sensors [33,34]. Except for the presence of EF-hands, CMLs do not possess any other functional domains and therefore do not possess any biochemical or enzymatic activity. Like CaMs, CMLs are also contain conserved Ca^2+^ binding signature sequences in their EF-hand domain. Sequence alignment showed the presence of a conserved D-x-D-x-D signature motif in the fourth EF-hand (Table 2) [19]. The conserved D-x-D-x-D motif amino acids of the EF-hand bind Ca^2+^ (Figure 2). The amino acids at the 14th, 16th and 18th position of the fourth EF-hand are conserved [19]. CaM proteins contain four conserved D-x-D motifs, one in each EF-hand, whereas CMLs contain only one D-x-D-x-D motif in the fourth EF-hand. This is the major molecular difference between the CaMs and CMLs. With the exception of the fourth EF-hand, none of the other three EF-hands in CMLs contain a conserved D-x-D motif [19]. Instead, the first EF-hand contains a conserved F-x_2_-F motif at the 5th and 8th position, a D-x_3_-D motif at the 9th and 13th position, a G at the 14th and an E at the 20th position [19]. The second EF-hand in CMLs also contain a conserved D-x3-D motif at the 13th and 17th position, a G at the 18th and E and F amino acids are conserved at the 24th and 25th position, respectively [19]. The third EF-hand contains conserved F, D and E amino acids at the 10th, 14th and 25th position, respectively [19]. In addition to the presence of a conserved D-x-D-x-D signature motif, the fourth EF-hand also contains an F-x-E-F conserved signature motif at the 23rd, 25th and 26th position [19]. The CMLs, relative to CaMs, contain an extended conserved signature motif of D amino acids. This suggests that the D-x-D motifs in CaMs have been substituted for D-x-D-x-D motifs in CMLs. The position of the conserved D amino acid in CaMs is different from CMLs.

Approximately 7.58% of the CML protein contain a conserved G amino acid at the 2nd position in the N-terminal region [19]. Co-translational and irreversible addition of myristic acid occurs in the N-terminal G amino acid leading to the myristoylation of proteins. Therefore, the presence of a conserved G amino acid at the 2nd position of the protein most likely results in the myristoylation of CMLs. The conserved myristoylation motifs present in the N-terminal end of CMLs are M-G-F, M-G-G and M-G-x (Table 3) [19]. This indicates that CMLs contain a signal sequence whereas CaMs do not. Myristoylation and palmitoylation events sometimes occur together and the absence of myristoylation may negate a palmitoylation event. Myristoylation event, however, can occur independently as well.

## 4. Calcineurin B-like (CBL)

Like *CaMs* and *CMLs*, plants possess a *CBL* gene family as well. An analysis of 38 plant species encompassing diverse genera indicates that plant genomes encode 2 to 14 *CBL* genes per genome [20]. The algal species, *Chlamydomonas reinhardtii* possesses the lowest number of *CBL* genes, whereas *Brassica rapa* contains the highest number of *CBL* genes among the plant species that have been thus far studied [20]. The *CBL* genes were not found in the algal species, *Coccomyxa subellipsoidea*, *Ostreococcus lucimarinus and Volvox carteri* [20]. The majority of *CBL* genes contain either six, seven, eight or nine introns, whereas only a few *CBL* genes are intronless [20]. The intronless *CBL* genes are present in *Micromonas pusilla* (*MpCBL2*) (algae) *Physcomitrella patens* (*PpCBL3-3*) (bryophytes) and *Selaginella moellendorffii* (*SmCBL5*) (pteridophytes) [20].

In contrast to the four EF-hands present in CaM and CML proteins, the CBL proteins contain only three EF-hands. The size of CBL proteins range from 119 amino acids (StCBL5) to 1015 amino acids (FvCBL4). The isoelectric point of CBL proteins are ranges from 3.94 (CreinCBL8) to 8.83 (MpCBL1) (Appendix A). Multiple sequence alignments of CBL proteins showed the presence of conserved sequence motifs. The first EF-hand contains conserved E amino acids at the 1st, 23rd and 24th position whereas a D amino acid is conserved at the 6th, 10th and 13th position. The Ca^2+^ binding D and E amino acids are conserved at the 3rd, 4th, 7th and 14th position [20]. The E amino acids are conserved at the 22nd, 25th and 36th position. The third EF-hand of CBL proteins contain a conserved D-D-x_2_-E motif at the 7th, 8th and 11th position and a E-E-x-D motif at the 19th, 20th and 22nd position [20]. The third EF-hand also contain a conserved D-x-E-E motif at 30th, 32nd and 33rd position [20]. The D and E amino acids in CBL proteins are responsible for binding Ca^2+^ ion (Figure 3). The conservation of calcium binding D and E amino acids in CBL proteins is more prominent in the 3rd EF-hand than in the 1st and 2nd EF-hands.

The positions of the conserved amino acids in CBL proteins are different from those found in CaMs and CMLs. The CBL proteins are classified into four groups namely group A, B, C and D [20]. Group D CBLs contain conserved D and E amino acids at the 16th, 17th and 18th (E-E-/D-P) positions at the N-terminal end of the protein (not in the EF-hand region). A conserved D/E-x-E/D motif is present upstream from the first EF-hand of group A CBL proteins [20]. In addition to these conserved regions, the N-terminal region of the CBL proteins contain a less conserved E/D-D-P-E-x_4_-E-x_6_-E motif and the C-terminal region contain a conserved P-S-F-V-F-x-S-E-V-D-E motif [20].

The N-terminal region of CBL proteins contain a conserved G amino acid at the 2^nd^ position that is a requisite for protein myristoylation (Table 3) [20]. In a few cases, the N-terminal G amino acid is conserved at the 7th position [20]. Additionally, the presence of a Cys (C) amino acid in the N-terminal region of CBL proteins may lead to protein palmitoylation. The C amino acid is conserved at the 4th position in group A CBLs, while it is conserved at the 3rd position in the N-terminal region in group D CBL proteins [20]. Group B CBLs do not contain an N-terminal C amino acid, suggesting that group B CBLs do not undergo protein myristoylation or palmitoylation [20]. The presence and absence of palmitoylation and myristoylation sites in different CBL proteins show, CBLs are involved in diverse cellular functions.

## 5. Calcium Dependent Protein Kinases (CPKs)

Calcium-dependent protein kinases are novel calcium sensors that are members of the serine/threonine protein kinase family. The number of *CPKs* genes in the plant genomes ranged from 2 (*Coccomyxa subellipsoidea* and *Micromonas pusilla*) to 53 (*Panicum virgatum*) [21]. *CreinCPK17-5* of *Chlamydomonas reinhardtii* is the largest reported *CPK* gene containing an ORF of 5940 nucleotides whereas *CpCPK2* of *Carica papaya* is the smallest *CPK* gene with an ORF of 693 nucleotides [21]. The presence of larger *CPK* genes in *Chlamydomonas reinhardtii* and *Physcomitrella patens* and the presence of smaller *CPK* genes in higher eukaryotes suggests, the evolution of eukaryotic *CPKs* genes are associated with loss of gene size [21]. The majority of *CPK* genes contain either 6, 7 or 8 introns and at least one *CPK* gene contains 11 introns from all of the species studied thus far, except for *Carica papaya, Ostreococcus lucimarinus and Ricinus communis* [21]. The phylogenetic analysis indicates the presence of four groups of plant *CPKs* that are designated as group A, B, C and D [21]. Furthermore, it appears that *CPK* genes originated from a common ancestor and that they have evolved very recently via gene duplication in which the genes possess similar or overlapping functions [21]. The presence of only four groups in the phylogeny of *CPK* genes indicates they have arisen through duplication.

The CPKs are evolutionarily conserved calcium binding proteins whose isoelectric ranges from 4.333 (OlCPK3) to 8.447 (SlCPK16) (Appendix A). Sequence alignments of CPKs indicate the presence of conserved D-x-D and D/E-E-L motifs in the EF-hands in both monocot and dicot plants [21]. Each of the four EF-hands contain at least one D-x-D motif. The species of algae, bryophytes and pteridophytes possess only two D-x-D motifs and only one D-x-x-D motif in the EF-hand [21]. The D-x-D motif in algae, bryophyte and pteridophyte present only in 3rd and 4th EF-hand [21]. The D-x-D motif is conserved at the 14th and 16th position of each EF-hand, whereas a D/E-E-L motif is conserved at the 24th, 25th and 26th position [21]. The D/E amino acids of the D-x-D and D/E-E-L motifs in CPKs are responsible for binding of Ca^2+^ ion (Figure 4). In addition to the D-x-D motif, the EF-hands in both monocot and dicot plants contain conserved E-E-I/x, D/E-E-L, D-Y-x-E-F, F-D-x-D, E-E-L, D-G-x-I and Y-x-E-F-x_2_-M-M motifs [21]. A conserved E-D-x_4_-A-F consensus sequence is found only in the CPKs of monocot plants [21]. In some cases, the conserved D-x-D motif is substituted by either x-D/E-L or Q-E-L (BdCPK30, OsCPK9, PvCPK30-1, PvCPK30-2, SbCPK30 and SiCPK30) or E-E-F-M (BdCPK7-2, BdCPK16-2, OsCPK3, OsCPK4, OsCPK8, PvCPK7-1, PvCPK8-2, PvCPK16-2, SbCPK8-1, SbCPK28, SiCPK7-1, SiCPK16, ZmCPK8-1, ZmCPK16-1/2 and ZmCPK32-1) motif [21]. In addition to the presence of conserved D-x-D and D/E-E-L motifs, the 1^st^ EF-hand in CPKs also contain a conserved E-E-I/x motif at the 1st, 2nd and 3rd position and a E-M-F motif at the 8th, 9th and 10th position [21]. The second EF-hand also contains a conserved E-x-E motif at the 3rd, 4th and 5th position, while the 3rd EF-hand contains a conserved x-E-D motif at the 3rd and 4th position. The 4th EF-hand in CPKs do not contain any conserved signature sequences at the same position [21]. Therefore, the N-terminal EF-hand pair (1st and 2nd EF-hands) possess the characteristic conserved signature sequences, while the C-terminal EF-hand pair (3rd and 4th EF-hands) do not contain any conserved signature sequences [21]. Instead, the 4th EF-hand contains conserved D/E amino acids at the 11th and 12th positions, respectively [21]. The presence of a conserved E-E-I/x motif at the 1st, 2nd and 3rd positions and an E-M-F motif at the 8th, 9th and 10th positions is specific to the 1st EF-hand. An E-x-E motif at the 3rd, 4th and 5th positions is specific to the 2nd EF-hand. An x-E-D motif at the 3rd and 4th positions is specific to the 3rd EF-hand and a conserved D/E amino acid at the 11th and 12th position is specific to the 4th EF-hand [21].

The first EF-hand in algae, bryophyte and pteridophyte contains a E-E-I/x motif at the 1st, 2nd and 3rd position whereas the first and fourth EF-hands contain a conserved D amino acid at the 14th position [21]. The second EF-hand in algae, bryophyte and pteridophyte possess conserved D and E amino acids at the 22nd and 25th position, respectively [21]. Differences in the number and position of conserved amino acids are the basis for variations in the allosteric properties of the Ca^2+^ binding and activation threshold of CPKs [37].

The sequence of the N-terminal region of CPKs is highly variable and dynamic while the kinase domain of CPKs contains conserved consensus sequences. The kinase domain in monocot and dicot plants contain the conserved consensus sequences C-x-G-G-E-L-x-D-R-I, H-R-D-L-K-P-E-N-F-L, D-x-V-G-S-x-Y-Y, A-P-E-V-L, D-V/I-W-S, G-V-I-x-Y-I-L-L, G-x-P-P-F-W, P-W-P-x-I-S, A-K-D-L-V and H-P-W. In contrast, the kinase domain in *Chlamydomonas reinhardtii, Volvox carteri, Micromonas pusilla, Ostreococcus lucimarinus and Physcomitrella patens* contain M-E-L-C-x-G-G-E-L-F, H-R-D-L-K-P-E-N-F-L, D-F-G-L-S-V/x, D-I-W-S-x-G-V and P-F-W conserved amino acid motifs [21]. The conserved consensus sequences D-x-V-G-S-x-Y-Y, G-V-I-x-Y-I-L-L, G-x-P-P-F-W, P-W-P-x-I-S, A-K-D-L-V and H-P-W are absent in algae, bryophyte and pteridophyte species [21]. The auto-inhibitory domain in CPKs is also highly conserved. The conserved consensus sequences present in the auto-inhibitory domain of monocot and dicot plants are K-P-L-D, F-S-A-M-N-K-L and A-L-x_2_-I-A, whereas in algae, bryophyte and pteridophyte the conserved domain contain A-M-N-K-L consensus sequence [21].

The CPKs contain putative myristoylation and palmitoylation sites [21,38]. Although the N-terminal domain in CPKs is variable, it does contain a conserved G amino acid at the second position and a C amino acid at either the 3rd, 4th, 5th or 6th position [21,39,40,41]. The N-terminal myristoylation supports protein-membrane attachment and protein-protein interactions [42]. Mutations in the N-terminal G amino acid suppress myristoylation and inhibit membrane association [41,43]. Sequence analysis indicates the presence of conserved myristoylation and palmitoylation sites in CPK [21]. The major myristoylation sites present in CPKs are M-G-N, M-G-C, M-G-L, M-G-G, M-G-S, M-G-I, M-G-Q, M-G-V and Q-F-G, while the major palmitoylation sites in CPKs are M-G-N-C, M-G-N-C-C, M-G-C, M-G-N-T-C-V, Q-F-G-T-T-F/Y-L/Q-C, M-G-N-C-C-R, M-G-L-C, M-G-G-C, M-G-N-N-C, M-G-S-C, M-G-N-S-C, M-G-I-C, M-G-N-C-N-A-C, M-G-Q-C, M-G-N-A-C, M-G-N-V-C, M-G-V-C, M-G-N-Q-C, M-G-N-C-N-T-C and M-E-L-C (Table 3) [21]. The N-terminal myristoylation of the G residue at the 2nd position and palmitoylation of the C residue at the 4th and 5th position has been experimentally validated for OsCPK2, a membrane bound CPK [41].

## 6. Interactome map of CaMs, CMLs, CBLs and CDPKs

All of the EF-hand containing protein performs the basic and common function of calcium binding to regulate calcium signaling event. However, it is indeed important to know whether all the calcium binding protein possess similar or diverse interacting partners. The cell should maintain the specificity of individual Ca^2+^ binding proteins to avoid unwanted crosstalk of calcium signaling event. The CaM protein interacts with diverse sets of proteins involved in calmodulin-dependent protein kinase activity. They include CREB (cAMP-response-element-binding protein) phosphorylation through the activation of CaMKK, ion channel transport, opioid signaling, CaM induced events, signal transduction, signaling by GPCR (Figure 5, Table 4 and Table 5) [44,45]. CREB is an important transcription factor involved in the activation of early genes and is phosphorylated by mitogen and stress activated kinase in response to activation of MAPK (mitogen activated protein kinase) [46]. The CaM1, CaM4, CaM5, CaM6 and CaM7 interacts with MPK8 (Figure 5). MAPK8 is involved in phosphorylation of diverse transcription factors. CaM3 interacts with CRLK1 (calcium/calmodulin-regulated receptor-like kinase 1) and AGL24 (MADS box protein) whereas CaM1 interacts with GAD (glutamate decarboxylase 1), LP1 (lipid transfer protein), AK1 (aspartokinase) and HY5 (basic leucine zipper) protein (Figure 5) [44,45]. The GAD protein catalyzes the α-decarboxylation of γ-aminobutyrate (GABA) contain a CaM binding domain at its C-terminal end [47]. Transient elevation of cytosolic Ca^2+^ due to various stresses lead to activation of GAD via CaM [47]. The LP1 protein act as anti-microbial, anti-fungal, anti-viral and enzymatic inhibitors, suggesting their role in pathogen defense mechanism [48]. Aspartokinase is involved in phosphorylation of amino acid aspartate that involved in the biosynthesis of amino acid methionine, lysine and threonine suggesting the involvement of CaM in biosynthesis of amino acids [49]. The HY5 transcription factor promotes photomorphogenesis in the presence of light. Pfam pathway show the involvement of CaM proteins in calcium and calmodulin binding, protein tyrosine kinase, ubiquitin-like autophagy and lipopolysaccharide pathway (Table 5).

CMLs were are majorly interact with ACOS5 (Acyl-CoA synthetase 5), TPC1 (two pore calcium channel 1), LAP6 (chalcone and stilbene synthase), TKPR1 (tetraketide alpha-pyrone reductase), NHX1 (sodium hydrogen exchanger 1), CCP3 (serine protease inhibitor 3), HSP (heat shock protein), DEK1 (calpain type cysteine protease), PHS2 (alpha glucan phosphorylase 2), LOS1 (ribosomal protein S5/elongation factor G/III/V family 1), WRKY (WRKY transcription factor), BAP1 (BON associated protein), SZF (zinc finger CCCH domain), RAD (UV excision repair protein), CIPK (CBL interacting protein kinase), ARP (DNA lyase) and other (Figure 6, Table 4) [44,45]. The 4-coumarate CoA-ligase-like and LAP6 are involved in phenylpropanoid metabolism and phenylpropanoid metabolism have diverse role in plant biotic and abiotic stress responses [50,51]. The TPC1 found in acidic organelle such as lysosome and vacuoles involved in transport of Ca^2+^ ion and regulate germination and stomatal movement [52,53]. The NHX1 protein is involved in regulation of cellular pH though the accumulation of Na^+^ in vacuoles [54]. Heat shock proteins act as molecular chaperons of plant immunity [55]. DEK1 is involved in development of epidermis and maintain adaxial/abaxial axis formation in developing leaves through the regulation of cell proliferation. PHS2 play important role in tolerance to abiotic stress in *A. thaliana* [56]. The RAD proteins are associated with chromatin in response to UV mediated DNA damage during S the phase [57]. WRKY40 is involved in plant biotic and abiotic stress responses through the induction of abscisic acid. WRKY46 is involved in brassinosteroid metabolism and modulate plant growth and drought tolerance [58,59]. BAP1 gene in A. thaliana act as general inhibitor of programmed cell death. The BON1 negatively regulate defense response gene SNC1 [60]. Overall, the interacting proteins are involved in nucleotide excision repair, neddylation, transport, cellular development and cellular response to biotic and abiotic stresses (Table 5) [44,45]. Pfam pathway also shows their involvement in WRKY DNA binding, EF-hand domain pair, TPR-repeats, Hsp90, ubiquitin pathway, chalcone and stilbene synthesis (Table 5) [44,45].

In the majority of cases CBLs are interact with CIPKs (CBL-interacting protein kinases) and regulate diverse functions including abscisic acid signaling and salt tolerance and [61,62]. The CIPKs which are the major interacting partner of CBLs are involved in immune response, response to salt, osmotic and abscisic acid stress, plant growth and development [61,62,63]. Wheat CIPK 25 is negatively regulate salt tolerance in the transgenic wheat [64]. In addition to interaction with CIPKs, CBLs are also interact with AKT1 (potassium channel 1), MTN1 (5′-methylthioadenosine/S-adenosylhomocysteine nucleosidase 1), SOS2 (CBL-interacting serine/threonine protein kinase 2), NHX1 (sodium hydrogen exchanger 1), SIP3 (CBL-interacting serine/threonine protein kinase 6), FKBP12 (peptidyl-prolyl cis trans isomerase), VSR2 (vacuolar sorting receptor 2), GLR2.8 (glutamate receptor 2.8), HAK5 (high affinity K^+^ transporter 5), TPC1 (two pore calcium channel protein 1), AVP1 (pyrophosphate-energized vacuolar membrane proton pump 1) (Figure 7, Table 4). MTN1 catalyzes irreversible breakage of glycosydic bond in 5-methylthioadenosine and S-adenosylhomocysteine to adenine, 5-methylthioribose and s-ribosylhomocysteine, thus maintaining AdoMet homeostasis and ethylene biosynthesis [65]. *A. thaliana* SOS2 is essential for intracellular Na^+^ and K^+^ homeostasis and regulate important role toward low K^+^ and high Na^+^ concentration [66]. FKBP12 protein is an important gene involved in physiologic regulator of cell cycle [67]. FKBPs are molecular chaperon and regulate diverse cellular process including adaptation to photosynthetic adaptations [68,69]. Glutamate receptor in plants is involved in light signal transduction and two pore calcium channel is actively involved in calcium signaling [60,70]. Vacuolar sorting receptors are involved in targeting of soluble cargo proteins to their respective destination [71]. Functional pathway shows the involvement of CBLs in calcium ion binding, potassium ion transmembrane transporter activity, ion gated channel activity and others (Table 5) [44,45].

CPKs are involved in diverse cellular function starting from cell signaling to stress tolerance in plants. They act as a hub of plant stress signaling and development [72]. The CPK genes are predominantly expressed in pollen where AtCPK17/AtCPK34, AtCPK11/AtCPK24 play role in growth of pollen tube [72]. AtCPK4, AtCPK5, AtCPK6 and AtCPK11 modulate pathogen associated molecular pattern [73]. AtCPK5 and AtCPK6 are involved in reactive oxygen species mediated cell death [74]. Interactome map generated from string database show the majorities of CPKs undergo cross interactions through the network of events (Figure 8, Table 4). However, in addition to the interaction with other CPKs, CPK4 interacts with ABF1 (abscisic acid responsive element-binding factor 1), RBOHD (respiratory burst oxidase homology D); CPK6 interact with OZS1 (C4-dicarboxylate transporter/malic acid transport protein) and ORP2A (oxysterol-binding protein-related protein 2A), CPK9 interacts with CRK1 (CDPK related kinase 1) and CPK19 interacts with F5M15.5 (catalase) (Figure 8, Table 4). The interactome map of CPK revealed that CPKs are involved in diverse cellular events involved in abscisic acid signaling, oxidative burst (early response to stress) and homeostasis of cellular oxidation through action of catalase [8,75].

## 7. Evolution of Calcium Signaling Events

The evolution of multi-cellular life forms from a single-celled ancestor is one of the most remarkable transitions that occurred during the evolution of life. Calcium ions may have a role as a “promoter” in the evolution of early forms of life [13]. Although, several different ionic molecules are present and available in the cytosol and on extracellular surfaces, Ca^2+^ dominated as the major cation for use as a universal intracellular signaling molecule [76,77]. This was most likely due to the properties of Ca^2+^ which possesses flexible coordination and rapid binding kinetics with biological molecules [78,79]. Additionally, when life arose in the ocean environment, sustaining cells in this oceanic environment would have been problematic since it contains high amount of Ca^2+^ and other minerals. Ca^2+^ ion enter into cells through different pumps and ion channels present in the cell membrane [80,81,82,83]. High concentrations of Ca^2+^ are cytotoxic [84,85,86]. Therefore, cells developed “tool kits” consisting of calcium sensing molecules that are capable of binding Ca^2+^ to maintain cellular homeostasis of calcium levels [9,87,88,89]. In addition to mitochondria and the endoplasmic reticulum, vacuoles in plant cells serve as storehouses for calcium ions [90,91,92,93,94,95]. Calcium sensing molecules, along with organelles to sequester calcium ions, provide a variety of sophisticated, mechanistic possibilities for regulating calcium homeostasis in cells [96].

The major EF-hand-containing calcium sensing molecules are calcium-dependent protein kinases, calmodulins, calmodulin-like and calcineurin B-likes proteins. The intra-genomic diversity of Ca^2+^ tool kit components has been reported to have emerged at bigger rate as compared to the evolution of other proteins [12]. Therefore, we constructed a phylogenetic tree in order to better understand various evolutionary aspects of EF-hand containing calcium signaling proteins (Figure 9). An analysis of the phylogenetic tree indicates that calcium sensing molecules have a polyphyletic origin. CMLs appear to be the most primitive calcium binding proteins followed by the evolution of CaMs, CBLs and CPKs. The phylogenetic tree of CPKs, CMLs, CaMs and CBLs is divided into five clusters. Cluster I contain CPKs (red) and CBLs (navy blue) and is divided into two sub-groups. Cluster I, in addition to CPKs and CBLs, also contains a few CaMs and CMLs. Cluster II contains CaMs (aqua) and CMLs (fuchsia) but no CPKs or CBLs. Cluster III contains only CMLs (fuchsia), while cluster IV contains CMLs and a CaM (VcCaM3). Cluster V contains CMLs and a CBL (CreinCBL9) (Figure 9). The presence of CMLs in all of the clusters and the absence of CBLs and CPKs in cluster II, III and IV indicates that CaMs, CBLs and CPKs evolved from CMLs and more specifically that CPK and CBLs evolved most recently.

A species tree was constructed to discern the duplication and loss events that had occurred in the evolution of EF-hands containing calcium-sensing molecules. Results indicated that the calcium-sensing molecules underwent duplication events during their evolution (Figure 10). Further duplication and loss analysis indicated that duplication of calcium-sensing genes preceded the loss events. Only a few calcium sensing genes that had been duplicated in the *A. thaliana* genome were had been lost in the *Oryza sativa* genome (Figure 10). These findings suggest that calcium sensing genes evolved in a lineage-specific manner. Lineage-specific evolution leads to a diversification in organismal complexity. The existence of a high number of CMLs and CPKs indicates that Ca^2+^ binding proteins associated with calcium signaling have expanded enormously during evolution [12,19,20,21]. The potential for diverse Ca^2+^ decoding mechanisms may have fostered the early evolution of calcium signaling molecules.

Ca^2+^-decoding protein families have greatly expanded and diversified in plants. An example of this is the existence of the large and diverse CML, CaM, CPK and CBL gene families. The presence of only one and three CaMs in *Eucalyptus grandis* (dicot) and *Coccomyxa subellipsoidea* (algae)*,* respectively, expanded up to 13 in *Brassica rapa* (dicot) and *Mimulus guttatus* (eudicot)*,* whereas the presence of two CMLs in *Coccomyxa subellipsoidea* (algae) and *Ostreococcus lucimarinus* (algae) expanded up to 47 in *A. thaliana* (dicot) (Table 1). Similarly, the absence of CBLs in *Coccomyxa subellipsoidea* (algae) contrasts with a total of 14 CBLs in *Brassica rapa* and the two CPKs in *Coccomyxa subellipsoidea* (algae) and *Ostreococcus lucimarinus* (algae) have expanded to 53 CPKs in *Panicum virgatum* (monocot). These examples reflect the evolutionary expansion and diversity of calcium-decoding molecules. Most unicellular eukaryotes, including algae, have a low calcium requirement for normal growth and development [12]. Therefore, they contain only a few numbers of CaMs, CMLs and CPKs in their genomes, while CBLs are absent in *Coccomyxa subellipsoidea*. Although several other EF-hand containing proteins are exist in plant cells, CMLs, CaMs, CBLs and CPKs account for more than one-third of all the EF-hand containing proteins present in plant genomes [12]. This suggests that, despite the hypothetical bottleneck in evolutionary events that occurs in complex, multicellular plants, CMLs, CaMs, CPKs and CBLs became diversified and gained multiple distinct functions. The complexity of Ca^2+^ signaling systems in plants is congruent with the increasing cellular and morphological complexity that has evolved in plants along with their ability to adapt to and live in a fluctuating and often harsh environment. The CPKs are quite abundant in algae and higher plants and they cluster with CBLs in phylogenetic trees. This suggests that CPKs and CBLs played prominent functional roles in the evolution of land plants. It seems that plants expanded the number of proteins within specific protein families during the evolution to foster functional diversification and specialization or that new functions evolved within existing protein families. The number of Ca^2+^-decoding protein families and the number of proteins within each family, was originally very low and subsequently expanded during evolution. The CMLs and CPKs families expanded dramatically and the increase in the number of family members is correlated with the increase in biochemical complexity that arose during plant evolution. The increase in organismal complexity was a driving force in the expansion and diversification of plant Ca^2+^ signaling systems where new functions had to be regulated by a limited number of sensors. This led to their expansion and functional diversification. The presence of conserved D-x-D motifs in the EF-hands of CaMs and CPKs suggest that the four primitive EF-hands of CaMs merged with protein kinase molecules to give rise to calcium-dependent protein kinases. The increase in CaM and CMLs gene numbers are correlated with the transition of plants from aquatic to terrestrial life forms [19]. The CaM proteins are highly conserved across all eukaryotes and CMLs are absent in the Unikonta super-group [19]. The acquisition of novel functions may have contributed to the ability of plants to adapt to the colonization of land habitats and the expansion and diversification of calcium-decoding proteins was certainly driven by selective pressures. The increase in the number of *CaM* and *CML* genes is correlated with the plant adaptation to a terrestrial environment (*Physcomitrella patens*, Bryophytes) and the subsequent organismal complexity of gymnosperms and angiosperms.

## 8. Conclusions

The genomic features of calcium binding proteins are quite conserved and interactome network reflect the involvement of CaMs, CMLs, CBLs and CPKs in diverse cellular events. More specifically CBLs are interacts with CIPKs and CPKs interacts with themselves. A complex set of evolutionary events have played an important role in shaping the Ca^2+^ signaling systems in plants. A genomic analysis of Ca^2+^ signaling system genes in plants provides a clearer picture of the evolution of the increased number of Ca^2+^ signaling proteins and their diverse roles in plant metabolism, physiology and morphology. The mapping of the duplication and loss events that have occurred in Ca^2+^ signaling sensor molecules provides clarification of the evolution of these gene families. The interactions of CPKs with other CPKs suggests, CPKs are most possibly co-regulated together and the presence of conserved molecular feature of D-x-D, D/E-E-LD motif as well as the presence of conserved D and E amino acid at different position across the EF-hand containing protein reflect their conserved evolutionary perspectives. Differential kinetic studies of calcium signaling associated enzymes can also be very useful for understanding the functional diversification of Ca^2+^ signaling cascades. Additionally, more research is needed to clarify the role of the chloroplast and mitochondria in the evolution of calcium-decoding molecules, as these organelles also play a critical role in Ca^2+^signaling. Functional characterization of conserved calcium binding motifs and their binding kinetics will be an excellent approach to better understand calcium signaling events in detail.

## Figures and Tables

**Figure 1 ijms-20-01476-f001:**
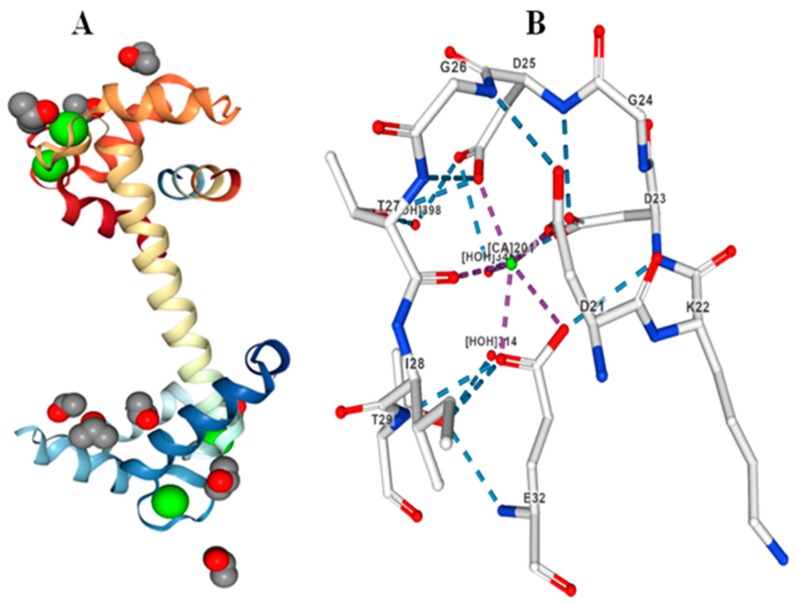
Molecular structure of CaM protein. (**A**) Three-dimensional structure of CaM protein with Ca^2+^ ligands bind to the EF-hands. (**B**) CaM protein binding to the Ca^2+^ ion in the D-x-D motif. Three water molecules (vertices -X) are present adjacent to the Ca^2+^ ion with strong hydrogen bonding. The molecular structure of CaM protein was elucidated using calmodulin protein 1UP5 as reported by Rupp et al., (1996) [28].

**Figure 2 ijms-20-01476-f002:**
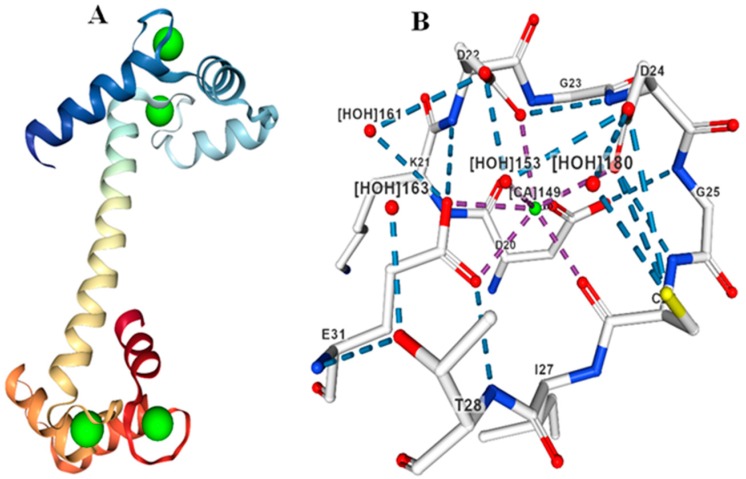
Molecular structure of CML protein. (**A**) Three-dimensional structure of CML protein with Ca^2+^ ligand binding to the EF-hands. (**B**) D amino acids binding with the Ca^2+^ ion in the EF-hand of CML protein. Four water molecules (vertices -X) are present adjacent to the Ca^2+^ ion with hydrogen bonding providing strong structural stability. The molecular structure of CML protein was created according to CML from protein 1UP5 from databank as reported by Rupp et al., (1996) [28].

**Figure 3 ijms-20-01476-f003:**
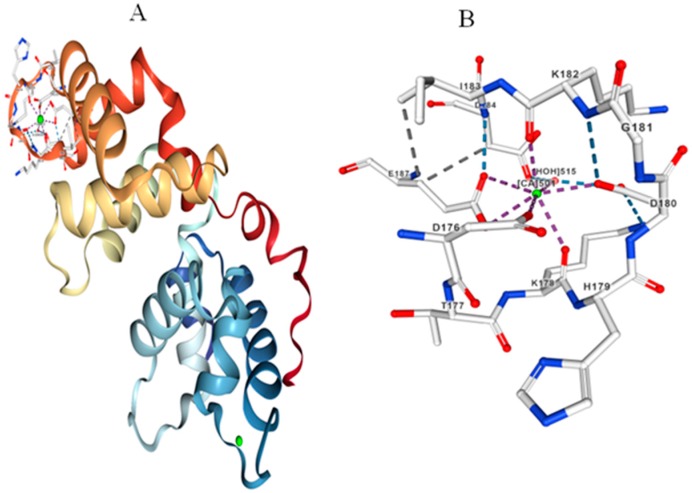
Molecular structure of CBL protein. (**A**) Three-dimensional structure of CBL protein with Ca^2+^ ligands binding to the EF-hands. (**B**) D amino acids binding to the to the Ca^2+^ ion in the EF-hand of CBL protein. One water molecule (vertices -X) provides structural stability to the Ca^2+^ ion in the EF-hand region of the CBL protein. The molecular structure of CPK protein was created according to *A. thaliana* CBL from protein databank as reported by Akaboshi et al., (2008) [35].

**Figure 4 ijms-20-01476-f004:**
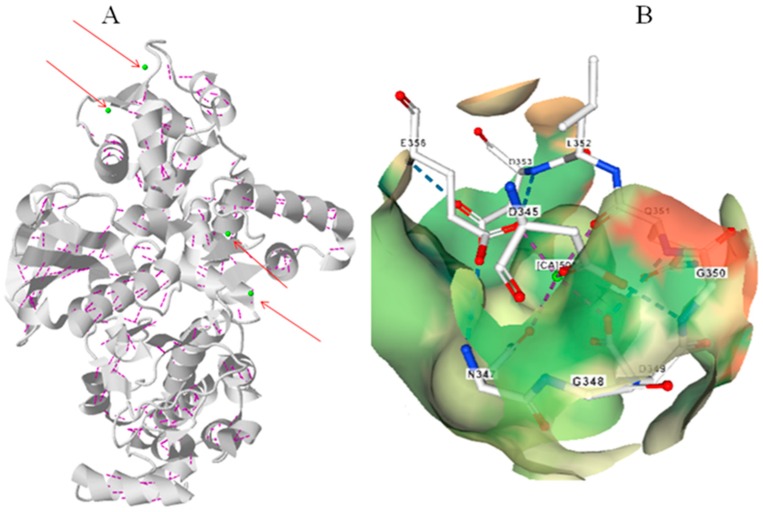
Molecular structure of CPK protein. (**A**) Three-dimensional structure of CBL protein with Ca^2+^ ligands marked in arrow. (**B**) D-x-D motif binding Ca^2+^ ion in the EF-hand of CPK protein. No water molecule was found to be associated with the Ca^2+^ ion in the EF-hand domain of CPK protein. The molecular structure of CPK protein was created according to *A. thaliana* CPK from protein databank as reported by Chandran et al., (2005) [36].

**Figure 5 ijms-20-01476-f005:**
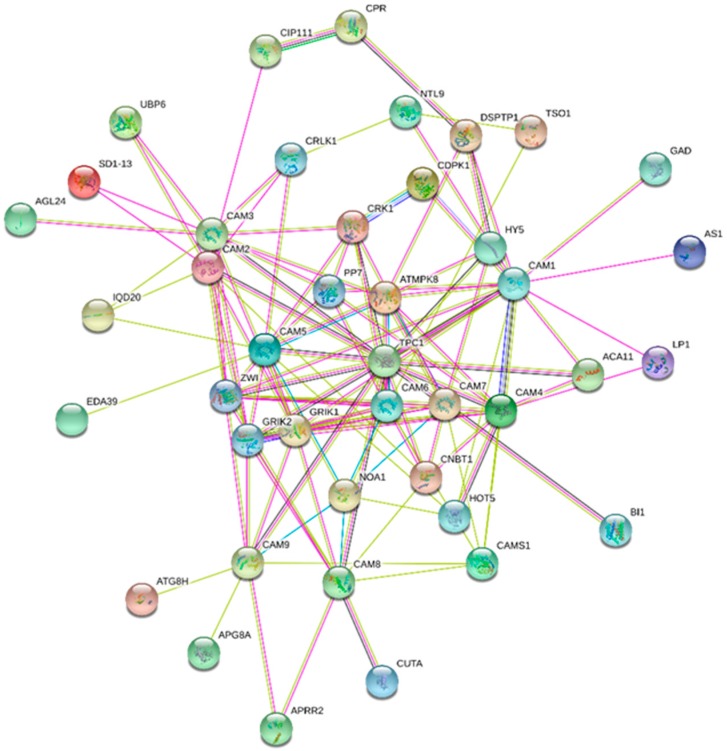
Interactome network of CaM proteins. The major interacting partners of CaM proteins are MPK8, TPC1, CDPK1, NOA1, UBP, HOT5 and others. The interactome network was constructed using the CaM proteins of model plant *A. thaliana*. String database was used to construct the interactome network [44,45].

**Figure 6 ijms-20-01476-f006:**
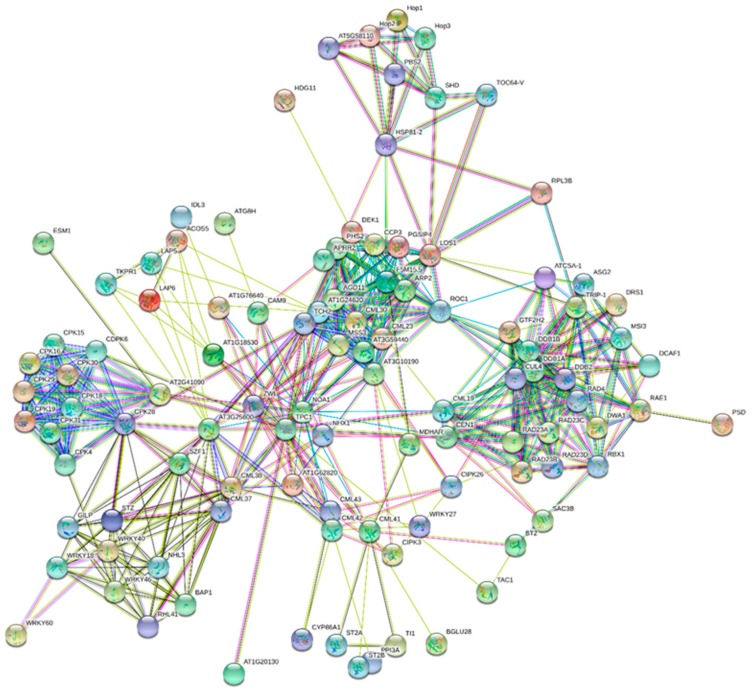
Interactome network of CML proteins. The major interacting partners of CML proteins are CLU4, MDHAR, WRKY transcription factor, CPKs, NHX, DDB1, RAD23A, RAD23C, RAD23D and others. The interactome network was constructed using the CML proteins of model plant *A. thaliana*. String database was used to construct the interactome network [44,45].

**Figure 7 ijms-20-01476-f007:**
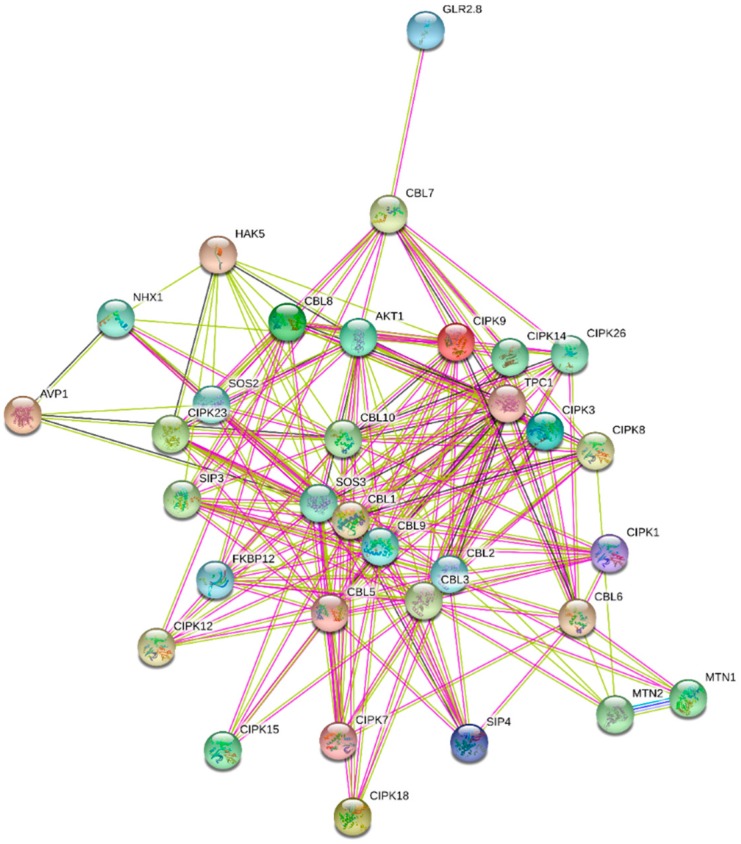
Interactome network of CBL proteins. The major interacting partners of CBL proteins are CIPKs. In addition, CBLs also interact with NHX1, MTN1, MTN2, SOS2 and other proteins. The interactome network was constructed using the CBL proteins of model plant *A. thaliana*. String database was used to construct the interactome network [44,45].

**Figure 8 ijms-20-01476-f008:**
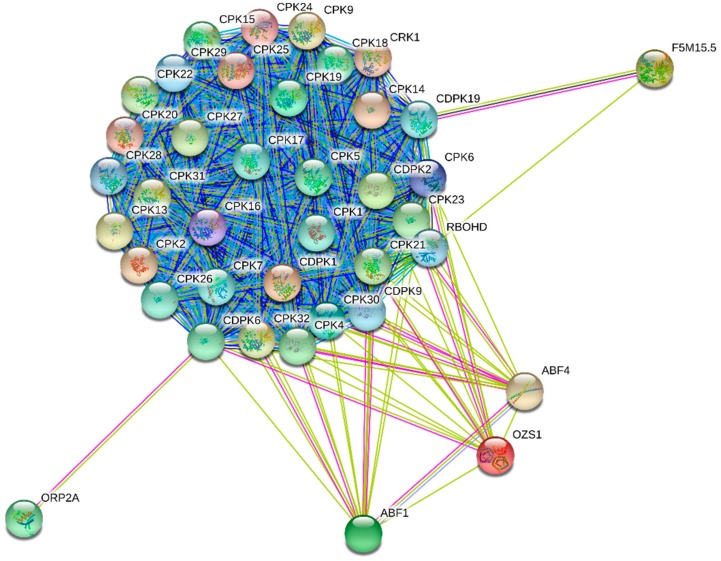
Interactome map of CPK proteins. The interactome map shows CPK proteins are interact with other CPKs proteins of the same protein family. However, CPKs are also interact with other proteins including ORP2A, RBOHD, ABF1, ABF4, OZS1 and F5M15.5. The interactome network was constructed using the CPK proteins of model plant *A. thaliana*. String database was used to construct the interactome network [44,45].

**Figure 9 ijms-20-01476-f009:**
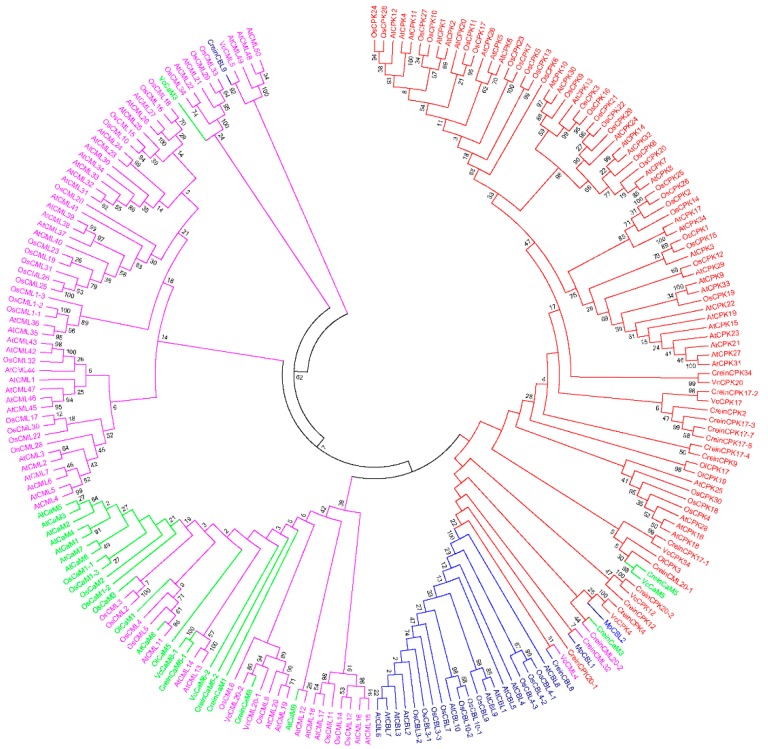
Phylogenetic tree of calcium sensor proteins. Phylogenetic tree of CaMs, CMLs, CBLs and CPKs resulted into five clusters. Cluster I contain CPKs (red) and CBLs (navy), cluster II contain CMLs (fuchsia) and CaMs (aqua), cluster III contain CMLs (fuchsia), cluster IV contain CaM (aqua) and CML (fuchsia) and cluster V contain CBL (aqua) and CML (fuchsia). Phylogenetic tress shows that CMLs are the primitive EF-hand containing calcium sensing proteins and evolved from monophyletic common ancestor [19]. Evolutions of CMLs were followed by the evolution of CaMs, CBLs and CPKs through gene duplication and subsequent expansion. Phylogenetic tree was constructed by maximum likelihood method and LG (Le & Gascuel) model using 1000 bootstrap replicates. Phylogenetic tree was constructed by MEGA6 software using the amino acid sequences of calcium sensor proteins.

**Figure 10 ijms-20-01476-f010:**
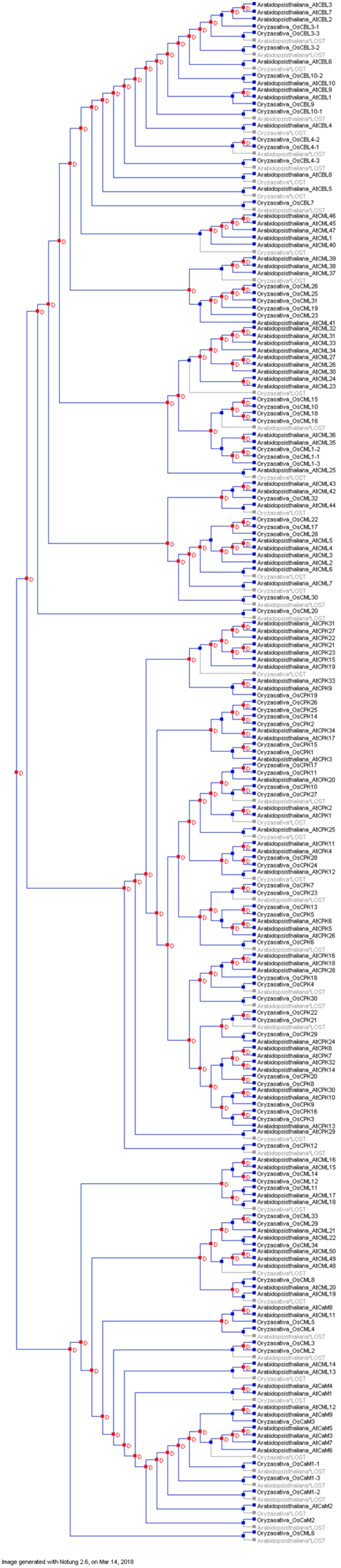
Duplication and loss event of calcium sensor proteins. A species tree was constructed to study the duplication and loss events of calcium binding sensor proteins. Result shows that, duplication event predominates the loss event during the evolution. Gene duplication events confirm the evolution of calcium sensing proteins through expansion. Duplicated genes in dicot plants are sometimes lost in the monocot lineage, suggesting the lineage specific evolution of calcium sensing genes. No conditional duplication was found during this study. The study was conducted using Notung software 2.9.

**Table 1 ijms-20-01476-t001:** Distribution of calcium binding protein CaMs (calmodulins), CMLs (calmodulin-like), CPKs (calcium dependent protein kinases) and CBLs (calcineurin B-like) in plants.

Sl. No.	Name of the Species	Taxonomy	Genome Size	Total No. of Protein Coding Genes	No. of CaMs	No. of CMLs	No. of CPKs	No. of CBLs
1	*Aquilegia coerulea*	Dicot	306.5	30023	5	21	16	5
2	*Arabidopsis thaliana*	Dicot	135	27416	9	47	34	10
3	*Brachypodium distachyon*	Monocot	272	34310	5	23	27	9
4	*Brassica rapa*	Dicot	283.8	40492	13	36	49	14
5	*Capsella rubella*	Dicot	134.8	26521	10	29	32	9
6	*Carica papaya*	Dicot	135	27332	5	15	15	4
7	*Chlamydomonas reinhardtii*	Algae	111.1	17741	6	3	14	2
8	*Citrus clementina*	Dicot	301.4	24533	8	19	26	7
9	*Citrus sinensis*	Dicot	319	25376	6	20	24	8
10	*Coccomyxa subellipsoidea*	Algae	49	9629	3	2	2	0
11	*Cucumis sativus*	Dicot	203	21491	6	21	18	7
12	*Eucalyptus grandis*	Dicot	691	36349	1	25	22	12
13	*Fragaria vesca*	Dicot	240	32831	5	19	14	6
14	*Glycine max*	Dicot	978	56044	6	27	41	9
15	*Gossypium raimondii*	Dicot	761.4	55294	6	30	40	13
16	*Linum usitatissimum*	Dicot	318.3	43471	11	21	47	12
17	*Malus domestica*	Dicot	881.3	63514	9	32	28	11
18	*Manihot esculenta*	Dicot	532.5	33033	9	22	26	9
19	*Medicago truncatula*	Dicot	360	50894	4	24	11	11
20	*Micromonas pusilla*	Algae	22	10660	5	8	2	3
21	*Mimulus guttatus*	Dicot	321.7	28140	13	19	25	9
22	*Oryza sativa*	Monocot	372	42189	5	33	30	11
23	*Ostreococcus lucimarinus*	Algae	13.2	7796	2	2	3	0
24	*Panicum virgatum*	Monocot	1358	102065	9	20	53	10
25	*Phaseolus vulgaris*	Dicot	537.2	27433	9	26	25	10
26	*Physcomitrella patens*	Bryophyte	480	32926	7	17	25	4
27	*Picea abies*	Pinophyta	1960	28354	9	15	11	13
28	*Populus trichocarpa*	Dicot	422.9	42950	8	26	28	11
29	*Prunus persica*	Dicot	225.7	26873	4	21	17	7
30	*Ricinus communis*	Dicot	400	31221	4	8	15	8
31	*Selaginella moellendorffii*	Pteridophyte	212.5	22273	6	11	9/11	4
32	*Setaria italica*	Monocot	405.7	34584	5	17	27	7
33	*Solanum lycopersicum*	Dicot	900	34727	9	27	28	11
34	*Solanum tuberosum*	Dicot	723	39028	5	27	21	12
35	*Sorghum bicolor*	Monocot	693.9	35490	8	22	28	8
36	*Thellungiella halophila*	Dicot	238.5	26351	10	27	31	9
37	*Theobroma cacao*	Dicot	346	29452	2	14	17	7
38	*Vitis vinifera*	Dicot	487	26346	5	13	17	9
39	*Volvox carteri*	Algae	131.2	14247	4	4	6	0
40	*Zea mays*	Monocot	2400	63540	8	21	47	9

**Table 2 ijms-20-01476-t002:** Conserved motifs of calcium binding protein CaMs, CMLs, CBLs and CPKs. In CaMs and CPKs D-x-D motifs are conserved at 14th and 16th position in each EF-hand.

EF-Hands	CaMs	CMLs	CBLs	CPKs
1st	D-x-D, E-x_2_-E	D-x_3_-D, F-x_2_-F	V-F-H-P-N	D-x-D, D-E-E-L, E-E-I, E-M-F
2nd	D-x-D, D-F-x-E-F	D-x_3_-D	D & E	D-x-D, D-E-E-L, E-x-E
3rd	D-x-D	D & E	E-E-x-D, D-D-x_2_-E	D-x-D, D-E-E-L, x-E-D
4th	D-x-D	D-x-D-x-D, F-x-E-F		D-x-D, D-E-E-L, D/E

**Table 3 ijms-20-01476-t003:** N-terminal myristoylation and palmitoylation sites of calcium binding proteins CaM, CML, CBL and CPKs. The presence of palmitoylation and myristoylation sites were studied using CSS palm2.0 software.

Calcium Binding Proteins	Myristoylation Sites	Palmitoylation Sites
CaM	Not present	Not present
CML	M-G-F, M-G-G, M-G-A	Not present
CBL	M-G-C	M-G-C, M-L-Q-C
CPK	MGN, MGC, QFG, MGL, MGS, MGI, MGQ, MGV,	M-G-N-C, M-G-N-C-C, M-G-C, M-G-N-T-C-V, Q-F-G-T-T-Y-L-C, M-G-N-C-C-R, M-G-L-C, M-G-G-C, M-G-N-N-C, M-G-S-C, M-G-N-S-C, Q-F-G-T-T-F-L-C, M-G-I-C, M-G-N-C-N-A-C, M-G-Q-C, M-G-N-A-C, Q-F-G-T-T-Y-Q-C, M-G-N-V-C, M-G-V-C, M-G-N-Q-C, Q-F-G-I-T-Y-L-C, M-G-N-C-N-T-C, M-E-L-C

**Table 4 ijms-20-01476-t004:** Table showing the interacting partners of CaMs, CMLs, CBLs and CPKs. CBL protein majorly interacts with CIPKs (CBL-interacting protein kinases) whereas majority of CPKs are interacts with CPKs themselves.

Gene Name	Interacting Proteins
**CaM**	
CaM8	APRR2, TPC1, At2g01210, NOA1, At2g20050, At1g73440, CUTA, GRIK2, GRIK1, CAMS1
CaM1	ZW1, CDPK1, NTL9, At2g01210, DSPTP1, CRCK, GAD, At5g62570, At5g28300, AS1
CaM2	UBP6, ZWI, At3g07670, At5g14260, CIP111, At1g24610, At5g53920, IQD20, SD1-13
CaM3	PP7, ZWI, UBP6, MPK8, TPC1, CRK1, SD1-13, AGL24, At3g07670, At5g14260
CaM5	MPK8, At2g19750, TPC1, EDA39, CRK1, NOA1, PP7, CRLK1, At2g02970
CaM4	HOT5, CNBT1, At4g33080, ACA11, TPC1, PP7, MPK8, At2g01820, LP1, CaM1
CaM7	At5g57110, MPK8, HY5, CNBT1, BI1, TPC1, At2g18750, NOA1, TSO1-like
CaM8	APRR2, TPC1, At2g01210, NOA1, At2g20050, At1g73440, CUTA, GRIK2, GRIK1, CAMS1
CaM9	APRR2, At1g73440, TPC1, ATG8H, APG8A, At2g27480, TSO1-like
**CML**	
CML15: AT1G18530	ACOS5, TPC1, LAP6, At2g27480, LAP5, GDSL, TKPR1, NHX1, CYP704B1
CML25: AT1G24620	CCP3, HSP81-3, ARP2, HSP81-4, DEK1, PHS2, LOS1, PGSIP4, HSP90.1
CML14: AT1G62820	CML43, CML42, TPC1, LRR, ZWI, NOA1, BIP2
CML23: AT1G66400	TCH2, MSS3, ARP2, DEK1, PHS2, LOS1, HSP81.3, HSP81.4, HSP90.1
CML39: AT1G76640	At1g54850, IDL3, TPC1, NOA1
CML38: AT1G76650	WRKY40, SZF1, STZ, BAP1, TPC1, RHL41,
CML30: AT2G15680	ARP2, DEK1, PHS2, LOS1, HSP81, ROC1
CML10: AT2G41090	At3g19100, CPK30, CDPK6, CPK31, CPK29, CPK16, CPK18, CPK4, CPK28, CPK15
CML5: AT2G43290	TCH2, CML23, ARP2, DEK1, PHS2, LOS1, At2g41410, At1g07940
CML3: AT3G07490	HSP70, ARP2, DEK1, PHS2, LOS1, PGSIP4, HSP81
CML36: AT3G10190	At1g62820, ARP2, DEK1, LOS1, HSP81, HSP90, CML30
CML16: AT3G25600	At4g27280, At3g10300, TPC1, At4g26470, SYP122, GILP, NHL3
CML20: AT3G50360	RAD4, RAD23B, RAD23A, RAD23D, RAD23C, DDB1B, DDB1A, GTF2H2, SAC3B
CML41: AT3G50770	At5g43260, TPC1, BGLU28, TI1, MDHAR, BT2, NOA1, ST2A
CML9: AT3G51920	APRR2, TPC1, ATG8H, APG8A, CDPK19,
CML4: AT3G59440	F5M15.5, ARP2, DEK1, PHS2, LOS1, HSP81, HSP90, ROC1
CML42: AT4G20780	TPC1, PPI3A, CYP86A
CML19: AT4G37010	RAD4, RAD23B, RAD23A, RAD23D, RAD23C, DDB2, SAC3B, At5g16090, CUL4, GTF2H2
CML37: AT5G42380	WRKY46, RHL41, BAP1, WRKY40, TPC1, GILP
CML43: AT5G44460	TPC1, CYP86A1, WRKY27, NOA1, CIPK26, CIPK3
**CBL**	
CBL1	CIPK23, AKT1, SOS2, CIPK1, CIPK7, At2g20050, CIPK15, CIPK8, CIPK26
CBL2	CIPK14, SIP4, At2g20050, CIPK23, At3g51390, SOS2, SIP3, CIPK12, CIPK18, MTN1
CBL3	CIPK23, At2g20050, SIP3, CIPK9, MTN1, CIPK1, MTN2, At3g51390, SOS2, CIPK14
CBL4	SOS2, SOS1, SIP3, NHX1, AKT1, At2g20050, At1g61575, CIPK8, CIPK14
CBL5	At2g20050, FKBP12, AKT1, CIPK1, CIPK23, SOS2, SIP4, CIPK14, TPC1, At3g59440
CBL6	VSR2, MTN1, FKBP12, CIPK9, CIPK1, TPC1, At4g10170
CBL7	GLR2.8, SOS2, CIPK3, CIPK26, CIPK9, FIBP12
CBL8	CIPK23, FKBP12, CIPK14, HAK5, TPC1, CIPK9, AKT1, SOS2, CIPK12
CBL9	CIPK23, AKT1, CIPK1, SIP3, CIPK8, SOS2, CIPK9, CIPK3, CIPK14
CBL10	SOS2, AKT1, CIPK23, SIP3, SOS1, FKBP12, CIPK8, TPC1, AVP1
**CPK**	
At5g24430	CPK1, At2g45300, CDPK1, CPK6, CPK9, At2g41090, CPK19, CPK24, CPK28, CPK32
CDPK1	CPK1, CPK4, CPK5, CPK6, CPK9, CPK28, CPK29, At5g34430, CPK32, At2g41090
CPK30	At2g41090, CRK1, CPK6, CPK9, CPK16, CPK18, CPK28, CDPK2, At5g24430, CPK23
CPK9	At2g41090, ABI2, CRK1, CPK4, At3g49370, At5g24430, CPK23, CPK28, CPK31, CPK32
CPK19	At2g41090, At5g24430, CRK1, CPK5, CPK9, CPK16, CPK21, CPK28, F5M15.5,
CPK7	CPK1, CPK4, CPK6, CPK9, CPK16, CPK18, CPK28, CPK29, CDPK1, At2g41090
CPK1	CDPK1, CDPK6, CPK28, CPK6, CPK4, CPK9, At2g41090, CPK32, CPK7, At5g24430
CPK18	At2g41090, CPK14, CPK19, CPK7, CDPK1, CPK5, CPK13, CPK32, CPK24, CPK30
CDPK6	At2g41090, ORP2A, CPK1, CPK9, CPK6, CDPK1, OZS1, CPK28, CPK18, CPK16
CPK15	At2g41090, CPK18, CPK16, CPK28, CPK24, CPK14, At3g49370, At5g24430, CPK30, CPK7
CPK4	ABF1, CPK28, CPK29, At2g41090, CPK1, CPK32, CPK6, CPK9, RBOHD
CPK31	At2g41090, CPK27, CRK1, At3g49370, CPK16, CPK18, CPK32, CPK25, CPK14, CPK13
CPK32	CPK28, ABF4, CPK6, CPK4, CDPK1, CPK9, CPK1, At2g41090, CPK9, CPK16
CPK2	At2g41090, CPK16, CPK19, CPK18, CPK5, At3g49370, At5g24430, CRK1, CPK1, At3g56760
CPK14	CPK24, CPK16, At2g41090, CPK20, CPK28, CPK18, CPK31, CPK29, CPK15, CPK19
CPK6	OZS1, CDPK1, CPK9, CPK28, CPK32, At5g24430, CPK29, CPK1, CPK4, CPK5

**Table 5 ijms-20-01476-t005:** Reactome, functional and Pfam pathway of calcium binding CaM, CML, CBL and CPK proteins. These proteins are involved in diverse cellular events. More specifically CMLs are involved in excision and DNA damage, CaM are involved in CaM induced signaling and CPKs are involved in abscisic acid signaling, oxidative burst and reactive oxygen species homeostasis. No reactome pathway was found for CBL proteins.

Reactome Pathway	Functional Pathway	Pfam Pathway	GO Component
**CML**			
DNA Repair	Calcium ion binding	EF-hand domain	GO:0005623→cell
DNA Damage Recognition in GG-NER	Binding	EF-hand domain pair	GO:0005622→intracellular
Formation of Incision Complex in GG-NER	Metal ion binding	EF-hand domain pair	GO:0044464→cell part
Nucleotide Excision Repair	Ion binding	EF hand	GO:0044424→intracellular part
Calmodulin induced events	Calmodulin-dependent protein kinase activity	EF hand	GO:0005634→nucleus
Ca-dependent events	Calcium-dependent protein serine/threonine kinase activity	XPC-binding domain	GO:0043229→intracellular organelle
CaM pathway	Damaged DNA binding	WD domain, G-beta repeat	GO:0043227→membrane-bounded organelle
DAG and IP3 signaling	Calmodulin binding	UBA/TS-N domain	GO:0005737→cytoplasm
Activation of CaMK IV	Protein binding	WRKY DNA -binding domain	GO:0043231→intracellular membrane-bounded organelle
Opioid Signaling	Hsp90 protein binding	TPR repeat	GO:0080008→Cul4-RING E3 ubiquitin ligase complex
G-protein mediated events	Organic cyclic compound binding	Kinase-like	GO:0031461→cullin-RING ubiquitin ligase complex
PLC beta mediated events	Heterocyclic compound binding	Tetratricopeptide repeat	GO:1990234→transferase complex
Intracellular signaling by second messengers	Proteasome binding	Sulfotransferase family	GO:0005829→cytosol
G2/M DNA damage checkpoint	Polyubiquitin modification-dependent protein binding	Ubiquitin family	GO:0005886→plasma membrane
G alpha (i) signaling events	Preprotein binding	Hsp90 protein	GO:0044444→cytoplasmic part
Signaling by GPCR	Ubiquitin binding	CPSF A subunit region	GO:0071944→cell periphery
GPCR downstream signaling	Protein serine/threonine kinase activity	3-Oxoacyl-[acyl-carrier-protein (ACP)] synthase III	GO:0005773→vacuole
Recruitment and ATM-mediated phosphorylation of repair and signaling proteins at DNA double strand breaks	Tetraketide alpha-pyrone synthase activity	Mono-functional DNA-alkylating methyl methanesulfonate N-term	GO:0016020→membrane
Neutrophil degranulation	Nucleic acid binding	Secreted protein acidic and rich in cysteine Ca binding region	GO:0032991→protein-containing complex
Immune System	Protein-containing complex binding	Tetratricopeptide repeat	GO:0034399→nuclear periphery
Post-translational protein modification	DNA binding	Protein kinase domain	GO:0098805→whole membrane
Josephin domain DUBs	Anion binding	Protein tyrosine kinase	GO:0005777→peroxisome
Metabolism of proteins	Purine ribonucleotide triphosphate binding	Chalcone and stilbene synthases, N-terminal domain	GO:0012505→endomembrane system
Dual Incision in GG-NER	Purine ribonucleotide binding	Tetratricopeptide repeat	
Signal Transduction	Serine-type endopeptidase inhibitor activity	Histidine kinase-, DNA gyrase B- and HSP90-like ATPase	
Neddylation	Drug binding	Sulfotransferase domain	
Formation of TC-NER Pre-Incision Complex	Small molecule binding	Tetratricopeptide repeat	
Dual incision in TC-NER	Nucleotide binding	Chalcone and stilbene synthases, C-terminal domain	
Gap-filling DNA repair synthesis and ligation in TC-NER	ATP binding	NAF domain	
HSP90 chaperone cycle for steroid hormone receptors (SHR)	Sulfotransferase activity	Histidine kinase-, DNA gyrase B- and HSP90-like ATPase	
N-glycan trimming in the ER and Calnexin/Calreticulin cycle	Transferase activity	Ubiquitin-2 like Rad60 SUMO-like	
Recognition of DNA damage by PCNA-containing replication complex	Enzyme binding	Lipopolysaccharide kinase (Kdo/WaaP) family	
Cellular responses to external stimuli	Sequence-specific DNA binding	C2H2-type zinc finger	
Cellular responses to stress	Transcription regulator activity		
Glycogen breakdown (glycogenolysis)	Catalytic activity, acting on a protein		
Protein methylation	Ras GTPase binding		
Cytosolic sulfonation of small molecules			
Asparagine N-linked glycosylation			
**CPK**			
Calmodulin induced events	Calmodulin-dependent protein kinase activity	EF-hand domain pair	GO:0005634 nucleus
Ca-dependent events	Calcium-dependent protein serine/threonine kinase activity	EF-hand domain pair	GO:0005886 Plasma membrane
CaM pathway	Calmodulin binding	EF hand	GO:0016020 membrane
DAG and IP3 signaling	Calcium ion binding	EF-hand domain	GO:0043231 Intracellular membrane bound organelle
Activation of CaMK IV	Protein binding	EF hand	GO:0005737 cytoplasm
G2/M DNA damage checkpoint	ATP binding	Protein kinase domain	GO:0071944 cell periphery
Opioid Signaling	Metal ion binding	Protein tyrosine kinase	GO:0005622 intracellular
G-protein mediated events	Heterocyclic compound binding	Lipopolysaccharide kinase (Kdo/WaaP) family	GO:0005623 cell
PLC beta mediated events	Organic cyclic compound binding	Secreted protein acidic and rich in cysteine Ca binding region	GO:0044464 cell part
Intracellular signaling by second messengers	Binding	Kinase-like	GO:0005829 cytosol
G alpha (i) signaling events	Catalytic activity	bZIP transcription factor	
Recruitment and ATM-mediated phosphorylation of repair and signaling proteins at DNA double strand breaks	Protein phosphatase binding	Basic region leucine zipper	
Signaling by GPCR	Peroxidase activity		
GPCR downstream signaling			
Signal Transduction			
**CaM**			
Neurotransmitter receptors and postsynaptic signal transmission	Calmodulin binding	Cytoskeletal-regulatory complex EF hand	GO:0005622 Intracellular
Post NMDA receptor activation events	Calcium ion binding	EF hand	GO:0005623 Cell
CREB phosphorylation through the activation of CaMKK	Binding	EF-hand domain pair	GO:0044464 Cell Part
Activation of NMDA receptors and postsynaptic events	Protein binding	EF-hand domain pair	GO:0044424 Intracellular part
Transmission across Chemical Synapses	Ion binding	EF hand	GO:0043229 Intracellular organelle
Neuronal System	Metal ion binding	EF-hand domain	GO:0005737 Cytoplasm
Ion channel transport	Protein serine/threonine kinase activity	EF-hand domain	GO:0043227 Membrane bound organelle
Opioid Signaling	Purine ribonucleotide binding	Lipopolysaccharide kinase (Kdo/WaaP) family	GO:0005874 Microtubule
Calmodulin induced events	Adenyl ribonucleotide binding	Autophagy protein Atg8 ubiquitin like	GO:0015630 Microtubule cytoskeleton
Ca-dependent events	Small molecule binding	Ubiquitin-like autophagy protein Apg12	GO:0043231 Intracellular membrane bound organelle
CaM pathway	Anion binding	Protein kinase domain	GO:0005634 Nucleus
G-protein mediated events	Drug binding	Protein tyrosine kinase	GO:0044446 Intracellular organelle part
PLC beta mediated events	Purine ribonucleoside triphosphate binding	Ion transport protein	GO:0000421 Autophagosome membrane
DAG and IP3 signaling	Organic cyclic compound binding	Kinase-like	GO:0005886 Plasma membrane
Signal Transduction	Heterocyclic compound binding		GO:0071944 Cell periphery
Activation of CaMK IV	ATP binding		GO:0005776 Autophagosome
G2/M DNA damage checkpoint	Catalytic activity, acting on a protein		GO:0043232 Intracellular non-membrane bound organelle
Intracellular signaling by second messengers	Voltage-gated cation channel activity		GO:0016020 Membrane
G alpha (i) signaling events	Calmodulin-dependent protein kinase activity		GO:0098805 Whole membrane
Recruitment and ATM-mediated phosphorylation of repair and signaling proteins at DNA double strand breaks	Calcium-dependent protein serine/threonine kinase activity		GO:0005774 Vacuolar membrane
Signaling by GPCR	Transcription regulatory region DNA binding		GO:0005773 Vauole
GPCR downstream signaling	Calcium ion transmembrane transporter activity		
	Metal ion transmembrane transporter activity		
	Identical protein binding		
	RNA polymerase II regulatory region sequence-specific DNA binding		
**CBL**			
NA	Calcium ion binding	NAF domain	GO:0005623 Cell
	Protein serine/threonine kinase activity	Kinase-like	GO:0044464 Cell part
	Ion binding	EF hand	GO:0005886 Plasma membrane
	Binding	Protein kinase domain	GO:0005622 Intracellular
	Catalytic activity, acting on a protein	Protein tyrosine kinase	GO:0000325 Plant-type vacuole
	Drug binding	EF-hand domain pair	GO:0071944 Cell periphery
	ATP binding	EF-hand domain pair	GO:0005737 Cytoplasm
	Metal ion transmembrane transporter activity	Phosphorylase superfamily	GO:0005774 Vacuolar membrane
	Potassium ion transmembrane transporter activity	Ion transport protein	GO:0016020 Membrane
	Cation channel activity	EF hand	GO:0005773 Vacuole
	Adenosylhomocysteine nucleosidase activity		GO:0009705 Plant-type vacuole membrane
	Methylthioadenosine nucleosidase activity		GO:0043231 Intracellular membrane bound organelle
	Metal ion binding		GO:0005634 Nucleus
	Ion gated channel activity		GO:0044444 Cytoplasmic part
	Sodium ion transmembrane transporter activity		
	Catalytic activity		
	Transmembrane transporter activity		
	Ligand-gated ion channel activity		
	Voltage-gated cation channel activity		
	Calcium channel activity		
	Potassium channel activity		
	Organic cyclic compound binding		
	Heterocyclic compound binding		
	Active transmembrane transporter activity		

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
