# Peer review of "Molecular Players of EF-hand Containing Calcium Signaling Event in Plants"

_ijms, 2019, doi:10.3390/ijms20061476_

Round 1
Reviewer 1 Report
The manuscript “Molecular players of EF-hand containing calcium signaling event in plants” submitted by Tapan Kumar Mohanta et al involves a review on plant EF-hand proteins. In general the manuscript is very good, including an extensive collection of acquired knowledge. Despite this feature, the manuscript can be substantially improved with a series of minor but essential amendments, such as:
1. Reduce sequential repetition in text. Often along the text the same information is repeated in 3-4 sequential sentences. It is highly recommended to avoid this practice. Please focus on the information transmitted in each paragraph.
2. Table 2 contains the number of EF-hand proteins of each type in different plants. Do numbers refer to isoforms? Are evidences of cation binding affinity variations among them?
3. In my opinion, this review gathers a very unique collection of information and ideas. However its strength is lost along the text. Probably reducing the amount of info to the essential core will permit to display the strength. For instance, giving the number of residues allows the estimation of the MW (then this info is not needed). However, including the pI may provide a classification into lineages as in other protein families.
4. Quality of figures must be improved. Those involving 3D structures have been stressed asymmetrically and balls appeared as ellipsoids. Also colors are too pale and boundaries are lost. In addition figures including trees are impossible to visualize properly.
Author Response
Dear reviewer,
Please find the response of your comments as attached file.
Thank you

Reviewer 2 Report
Comments on “Molecular Players of EF-hand Containing Calcium Signaling Event in Plants” by Mohanta et al.
This review manuscript discusses the proteins with E-F hand motif in the context of calcium signalling in plant cells. It covers protein structure, interactome, and evolution of calcium signalling.
While the manuscript has properly discussed some important aspects of the four main types of Ca2+ sensor proteins, it has not been written properly. In particular, throughout the manuscript, there are many broken sentences, incorrect grammar or inaccurate descriptions. Consequently, the readability of the manuscript is poor. Moreover, the discussion/description has not been appropriately organised. Therefore, I suggest that the manuscript should be thoroughly revised.
1. Writing style
Please revise the manuscript thoroughly. I have listed several examples below, but in general the manuscript is written poorly.
Line 25: subsequently the four later four conserved EF-hand;
Line 402: CMLs were are majorly interacts with ACOS5;
Line 418: CBLs are majorly interacts with CIPKs;
….
2. Connection between sections
Sections 2-5 mainly discuss the structure of the four main types of Ca2+ sensor proteins in plants. Sections 6 and 7 mainly discuss interactome and evolution of calcium signalling. More discussion/description is required to connect those sections.
3. Conclusion Section
Conclusion section should be expanded to discuss how this review manuscript, which covers some important aspects of the four main types of Ca2+ sensor proteins, has helped better understand calcium signalling in plants. For example, the implications of different numbers of calcium-binding domains of the four main types of Ca2+ sensor proteins in calcium signalling could be further discussed.
4. Presentation
In the manuscript that I can access, Figure 10 is not readable (fonts are too small). I was not able to see Supplementary Figures, so I am unable to make comments on those figures.
Author Response
Dear reviewer,
Kindly find the response of your comments as attached file.
Thanks

Round 2
Reviewer 2 Report
Comments on the revised version of “Molecular Players of EF-hand Containing Calcium Signaling Event in Plants” by Mohanta et al.
Although the cover letter states that “the manuscript was edited by professional English editing services”, the English used in this manuscript is still incorrect. Therefore, its readability is still poor. My suggestion is that the authors should revise the manuscript more carefully.
Some examples are listed below:
Line 405: CMLs were are majorly interacts with ACOS5 (4-coumarate CoA-ligase-like;
Line 434: CBLs are majorly interacts with CIPKs (CBL-interacting protein kinase;
Line 438: In addition to interaction with CIPKs, CBLs are also interacts with;
Line: 447: A. thaliana SOS2 essential for intracellular Na+ and K+ homeostasis and important;
Line 458: they are also interact with NHX1;
Line 476: The interactome map shows CPKs are interact with CPKs;
Line 477: CPKs are also interacts with ORP2A,
………
Author Response
Dear reviewer,
Greetings
Many thanks for pinpointing these grammatical mistakes. Authors have gone through the manuscript again and corrected the necessary mistakes related to English typos and grammars.
Now we have addressed all of your suggestions. The modified parts in the manuscript marked in red.
Thank you again for providing your precious time to improve our manuscript in a short span of time.
Authors are like to appreciate you for your effort.
Have a nice time.
Sincerely
Response to reviewer comments
Line 405: CMLs were are majorly interacts with ACOS5 (4-coumarate CoA-ligase-like;
Response: Corrected
Line 434: CBLs are majorly interacts with CIPKs (CBL-interacting protein kinase;
Response: Corrected
Line 438: In addition to interaction with CIPKs, CBLs are also interacts with;
Response: Corrected
Line: 447: A. thaliana SOS2 essential for intracellular Na+ and K+ homeostasis and important;
Response: corrected
Line 458: they are also interact with NHX1;
Response: corrected
Line 476: The interactome map shows CPKs are interact with CPKs;
Response: corrected
Line 477: CPKs are also interacts with ORP2A,
Response: corrected